# TOWARDS EFFICIENT TRACE ESTIMATION FOR OPTIMAL TRANSPORT IN DOMAIN ADAPTATION

## ABSTRACT

We improve the efficiency of optimal transport problems with Laplacian regularization in domain adaptation for large-scale data by utilizing Hutchinson's trace estimator, a classical method for approximating the trace of a matrix which to the best of our knowledge has not been used in this context. This approach significantly streamlines the computational complexity of the Laplacian regularization term with respect to the sample size $n$, improving the time from $O(n^3)$ to $O(n^2)$ by converting large-scale matrix multiplications into more manageable matrix-vector multiplication queries. In our experiments, we employed Hutch++, a more efficient variant of Hutchinson's method. Empirical validations confirm our method's efficiency, achieving an average accuracy within 1% of the original algorithm with 80% of its computational time, and maintaining an average accuracy within 3.25% in only half the time. Moreover, the integrated stochastic perturbations mitigate overfitting, enhancing average accuracy under certain conditions.

## 1 INTRODUCTION

Domain Adaptation (DA) is a field in machine learning where the goal is to adapt the model trained on one domain to work well on a different but related domain. DA is critical for enhancing model generalization across diverse data distributions (Farahani et al., 2021). The distinct challenges in data acquisition and annotation, including dataset imbalance and inaccuracies, emphasize the significance of developing robust DA techniques. DA aims to counter the adverse effects of data shift by aligning models to the intricate variations in different data distributions. This is essential for optimizing performance in fields such as computer vision, natural language processing, and recommendation systems. Numerous methods address different aspects of the adaptation process in DA, such as feature selection techniques (Li et al., 2016), re-weighting strategies (Ringwald & Stiefelhagen, 2021), subspace mapping (Fernando et al., 2014), deep learning models (Kang et al., 2019). These varied approaches, each addressing distinct challenges in DA, collectively contribute to advancing model generalization and methodological refinement in DA settings.

In this context, Optimal Transport (OT) as a DA technique has emerged prominently, providing a sophisticated mathematical framework to meticulously align source and target domains by gauging the similarity between distributions (Courty et al., 2017). To preserve geometric structure among samples, a widely adopted approach involves integrating a Laplacian regularization term with conventional OT problems (Ferradans et al., 2014). By utilizing a similarity matrix to characterize the resemblance between samples, the Laplacian regularization term ensures that similar samples maintain their similarity post-transport. Laplacian regularized OT is notably leveraged in DA, exemplifying its capability to attain high accuracy (Courty et al., 2017). However, OT methods are generally more computationally intensive than non-OT algorithms such as Joint Distribution Adaptation (JDA), due to the necessity to solve complex optimization problems, often involving large-scale linear programming. This poses challenges in enhancing its computational efficiency in scenarios involving large-scale data or limited computational resources.

Addressing this, we observe that the Laplacian regularization term can be represented as the trace of a series of matrix products. However, efficient trace estimation in this context remains underexplored, limiting the scalability of Laplacian regularized OT methods in real-world scenarios. To overcome this limitation, this paper explores the use of Hutchinson's trace estimator for efficiently approximating the Laplacian regularization term, offering a solution to the computational challenges posed by OT.

The main contributions of this paper are:

- We introduce the utilization of Hutchinson's trace estimator for approximating the trace of a matrix in the Laplacian regularization term, and implemented its enhanced variant, Hutch++. This innovation significantly refines the computational complexity associated with the Laplacian regularization term in relation to the sample size $n$, improving the time complexity from $O(n^3)$ to $O(n^2)$.

- We leverage the query number parameter to establish a sophisticated trade-off between computational time and accuracy, catering to various real-world efficiency needs. Inherently incorporating stochastic perturbations, our method not only mitigates overfitting and demonstrates robustness to perturbations but also surpasses the original algorithm in average accuracy under certain conditions.

- Our empirical tests validate the effectiveness of our approach in achieving competitive accuracy with a much lower computational requirement. Empirical validations substantiate the efficiency of our method, realizing an average accuracy within 1% of the original algorithm at 80% of its computational time, and preserving an average accuracy within 3.25% in merely half the time.

## 2  RELATED WORK

**Applications of Optimal Transport.** Originating from the work of Gaspard Monge in 1781, OT has found extensive applications across various domains. It has been crucial in systems and control for addressing challenges such as spacecraft landing and thermal noise mitigation on resonators (Chen et al., 2021). Its application in image classification and object recognition is notable, with the development of Optimal Transfer GAN, which has established benchmarks in image generation tasks due to its stability (Salimans et al., 2018). In biomedical research, OT-driven tools like TrajectoryNet have elucidated cell dynamics in single-cell RNA sequencing data (Tong et al., 2020). Moreover, OT has facilitated advancements in image processing tasks such as color normalization and image segmentation (Papadakis, 2015).

**Efficient Computation Methods for Large Matrices.** The focus on efficient computational methods is pivotal in addressing challenges associated with large-scale matrix multiplication in trace functions. Strassen's algorithm (Huss-Lederman et al., 1996) significantly reduced the computational complexity of matrix multiplication methods. The advent of randomized decomposition algorithms (Erichson et al., 2019) has been fundamental in economizing computational resources for large-scale matrix decompositions and low-rank approximations. Stochastic gradient descent methods (ichi Amari, 1993) in machine learning have optimized computations, effectively mitigating the computational load of large-scale matrices. Distributed computing frameworks (Zhang et al., 2012) have been instrumental in enabling the scalable processing of large matrices. These methods have bolstered the efficient management of large matrices across various applications.

## 3  PRELIMINARIES

### 3.1  OPTIMAL TRANSPORT

OT explores efficient mass transference from one configuration to another, finding applications in various domains such as mathematics, economics, and machine learning. For conciseness, we discuss the Monge and Kantorovich formulations in brief below.

**Monge Problem:** Introduced by Monge in 1781, this problem seeks an optimal map $T : \Omega \rightarrow \Omega'$ to transport mass from a measure $\mu$ on $\Omega$ to $\nu$ on $\Omega'$, minimizing the cost $C(T) = \int_\Omega c(x, T(x)) \, d\mu(x)$ subject to $T_{\#}\mu = \nu$. Here, $c : \Omega \times \Omega' \rightarrow \mathbb{R}$ is the given cost function.

**Kantorovich Problem:** A generalized and more flexible approach than the Monge Problem, this problem considers transport plans, $\gamma$, a joint probability measure on $\Omega \times \Omega'$ with marginals $\mu$ and $\nu$, minimizing the cost $C(\gamma) = \int_{\Omega \times \Omega'} c(x, y) \, d\gamma(x, y)$ over all $\gamma$ in $\Pi(\mu, \nu)$. This formulation allows for a more flexible coupling between $\mu$ and $\nu$ compared to the deterministic mappings in the Monge framework.

### 3.2 Discrete Optimal Transport and Domain Adaptation

Moving from the continuous setting of OT, we focus on its discrete counterpart in DA. DA is pivotal when transferring knowledge from a labeled source domain to a related, unlabeled target domain. In our following presentation, we adopt the notation of Courty et al. (2017).

Given empirical distributions $\mu_s$ and $\mu_t$ of the source and target domains, respectively, the discrete version seeks a transport plan $\gamma \in \mathbb{R}^{n_s \times n_t}$ minimizing total transport cost under mass conservation constraints, akin to the Kantorovich formulation. The admissible transport plans are characterized by $\mathcal{B} = \{\gamma \in (\mathbb{R}_{\geq 0})^{n_s \times n_t} \mid \gamma 1_{n_t} = \mathbf{p^s}, \gamma^T 1_{n_s} = \mathbf{p^t}\}$, and the optimal $\gamma_0$ is obtained by solving $\gamma_0 = \underset{\gamma \in \mathcal{B}}{\mathrm{argmin}} \langle \gamma, C \rangle_F$,

with $C \in \mathbb{R}^{n_s \times n_t}$ being the cost matrix, $\langle \cdot, \cdot \rangle_F$ denoting the Frobenius inner product.

In the above, $\mu_s = \sum_{i=1}^{n_s} p_i^s \delta_{x_i^s}$ and $\mu_t = \sum_{i=1}^{n_t} p_i^t \delta_{x_i^t}$, where $p_i^s$ and $p_i^t$ are the weights of the $i$-th source and target samples, and $\delta_{x_i^s}$ and $\delta_{x_i^t}$ are the Dirac delta functions centered at the samples $x_i^s$ and $x_i^t$.

In DA, $\gamma_0$ can guide the adaptation of models from the source to the target domain, often by re-weighting the source samples according to $\gamma_0$ during model training. The cost matrix $C$, usually defined by the squared Euclidean distance $C(i, j) = \left\| x_i^s - x_j^t \right\|_2^2$, measures the cost of transporting samples between domains.

### 3.3 Laplacian Regularization

Regularization techniques have played a pivotal role in reducing overfitting. In DA, an extension of the OT problem incorporates class label information into the transport plan, achieved by introducing regularization terms into the objective function, enabling the transport plan to align with the geometric or semantic structures within and across domains. Specifically, we integrate both source domain structure and class-based regularization terms into the OT problem, modeled as Courty et al. (2017): $\min_{\gamma \in \mathcal{B}} \{\langle \gamma, C \rangle_F + \lambda \Omega_s(\gamma) + \eta \Omega_c(\gamma)\}$, where $\lambda$ and $\eta$ are regularization parameters and $\Omega_s(\gamma) = \sum_{i,j} \gamma(i, j) \log \gamma(i, j)$.

The class-based term, $\Omega_c(\gamma)$, minimizes the discrepancy between same-class samples and allows flexibility between different classes. It is realized through a Laplacian regularization:

$$\Omega_c(\gamma) = \frac{1}{n_s^2} \sum_{i,j} S_s(i, j) \|\hat{x}_{s_i} - \hat{x}_{s_j}\|_2^2. \tag{1}$$

Here, $S_s$ denotes the source domain sample similarity matrix, with $S^s(i, j) = 0$ for $y_i^s \neq y_j^s$, ensuring class consistency by sparsifying similarities for different classes. $\hat{x}_{s_i}$ denotes the transported source sample $x_{s_i}$.

With the formula of barycentric mapping $\hat{X}_s = n_s \gamma X_t$ under uniform marginals, the class-based term can be represented concisely through matrix operations as:

$$\Omega_c(\gamma) = \mathrm{Tr}(X_t^T \gamma^T L_s \gamma X_t), \tag{2}$$

where $L_s = \mathrm{diag}(S_s 1) - S_s$ is the Laplacian of graph $S_s$. $X_s$ and $X_t$ are the matrices consisting of the row vectors of all the elements of the source domain and target domain respectively, with $\hat{X}_s$ the matrix of

transported source domain vectors. $\text{Tr}(\cdot)$ is the matrix trace. The core objective of our study is the refinement of computational strategies for $\Omega_c(\gamma)$.

### 3.4 Portraying Similarities

In Equation 1, $S_s(i, j)$ quantifies the similarity between the $i^{th}$ and $j^{th}$ samples. Various methods exist to characterize the similarity. A straightforward approach is to employ the k-Nearest Neighbors (k-NN) algorithm to generate $S_s$, under the assumption that neighboring samples are similar. Alternatively, a more sophisticated strategy involves utilizing kernel functions, which are capable of quantifying the distance between samples, to compute the elements $S_s(i, j)$ of the similarity matrix. The *Gauss kernel* is expressed as $K(\boldsymbol{x}, \boldsymbol{x}') = \exp(-\|\boldsymbol{x} - \boldsymbol{x}'\|^2/(2\sigma^2))$, where $\sigma$ is the scale parameter. The *Matérn kernel*, introducing an additional smoothness parameter $\nu$, is formulated as $K(\boldsymbol{x}, \boldsymbol{x}') = (2^{1-\nu}/\Gamma(\nu))(\sqrt{2\nu}\|\boldsymbol{x} - \boldsymbol{x}'\|/l)^\nu K_\nu(\sqrt{2\nu}\|\boldsymbol{x} - \boldsymbol{x}'\|/l)$, with $l$ as the scale parameter, where $\Gamma$ is the gamma function and $K_\nu$ is the modified Bessel function of the second kind. The *neural network kernel*, inspired by single-layer perceptrons, is given by $K(\boldsymbol{x}, \boldsymbol{x}') = (2/\pi) \arcsin((2\boldsymbol{x}^T\Sigma\boldsymbol{x}')/\sqrt{(1 + 2\boldsymbol{x}^T\Sigma\boldsymbol{x})(1 + 2\boldsymbol{x}'^T\Sigma\boldsymbol{x}')})$, where $\Sigma$ is a matrix parameter (Williams, 1998). When performing Laplacian regularized OT, we choose the one with the best accuracy among the above methods.

## 4 Optimizing Laplacian Regularization

In this section, we aim to optimize the computation of Equation (2) discussed at the end of the previous section. We employ Hutchinson's trace estimator, a class of stochastic methods for estimating the trace in Equation (2). The impacts of these strategies are empirically evaluated in Section 5.

### 4.1 Hutchinson's Trace Estimator

Hutchinson's Trace Estimator is a stochastic method for estimating the trace of a matrix. The original paper Hutchinson (1989) has the following result: let $A$ be an $n \times n$ matrix. Let $z$ be a random vector whose entries are i.i.d. Rademacher random variables, i.e., $\Pr(z_i = \pm 1) = 1/2$. Then $z^T A z$ is an unbiased estimator of $\text{tr}(A)$, that is, $\mathbb{E}(z^T A z) = \text{tr}(A)$ and $\text{Var}(z^T A z) = 2\|A\|_F^2 - \sum_{i=1}^n A_{ii}^2$, where $\|A\|_F$ denotes the Frobenius norm of $A$. We represent Hutchinson's trace estimator as $H_M = \frac{1}{M} \sum_{i=1}^M z_i^T A z_i$, where the $z_i$ denote $M$ independent random vectors with entries that are i.i.d. Rademacher random variables. Girard (1987) validated that this estimator is equally effective when the entries are i.i.d. standard normal variables.

Hutchinson's method is instrumental for estimating the trace of a matrix $A$, avoiding its explicit computation and requiring only the knowledge of the products of $A$ with specific vectors. To elaborate, for computing $z^T A z$, knowing $Az$ is sufficient, with subsequent multiplication by $z^T$. This is particularly valuable when $A$ arises from complex operations on other large matrices, making querying the product of $A$ and a vector more time-efficient. For instance, in practice computing the series of matrices within the trace function in Equation (2) uses $O(n^3)$ time, considering multiple matrix multiplications of order $n$, where $n$ is the sample size. However, to query the product of these matrices with a vector $z$, we start with $z$ and sequentially multiply it by each matrix, updating the vector for the next matrix multiplication, all while maintaining $O(n^2)$ complexity. Explicitly, if we denote $(.,.)$ as the matrix-vector product, then the estimator for Equation (2) can be represented as $\Omega_c^M(\gamma) = \frac{1}{M} \sum_{i=1}^M (z_i^T, (X_t^T, (\gamma^T, (L_s, (\gamma, (X_t, z_i))))))$. This becomes advantageous when the number of queries $M$ is much less than the matrix dimensions. Additionally, the incorporation of Hutchinson's method not only enhances efficiency but also introduces random perturbations, serving as a measure against overfitting, a claim corroborated in Section 5.3.

## 4.2 Hutch++: Reducing Variance of Hutchinson's Estimation

Following the introduction of Hutchinson's trace estimator, we delve into its enhanced variant, Hutch++, proposed in Meyer et al.. Hutch++ is an algorithm tailored for optimal estimation of the trace of positive semi-definite (PSD) matrices by matrix-vector multiplication queries. The prowess of Hutch++ lies in its ability to compute a $(1\pm\varepsilon)$ approximation to $\mathrm{tr}(A)$ for any PSD matrix $A$ using merely $O(1/\varepsilon)$ matrix-vector multiplication queries. This is a marked improvement over Hutchinson's estimator that requires $O(1/\varepsilon^2)$ matrix-vector products. Furthermore, within a logarithmic factor, Hutch++ possesses optimal complexity among all matrix-vector query algorithms. Hutch++ is a variance-reduced adaptation of Hutchinson's estimator and is extensively used in our experiments.

# 5 Numerical Experiments

Our experimental platform is implemented on a Tencent Cloud server, equipped with a 16-core Intel Xeon Cascade Lake 6231C CPU. To assess performance, we employ a 1-Nearest Neighbor (1-NN) classifier to evaluate the accuracy, using processor time as the primary metric to evaluate the time efficacy. We leverage two benchmark datasets containing handwritten digits: MNIST and USPS. To align with the dimensionality of USPS, we utilize PCA to reduce the dimensionality of the MNIST datasets to $16 \times 16$. For convenience, the datasets are truncated, retaining only 1500 samples. Unless otherwise noted, each data point in the charts and data we present in this paper is the average result of 100 replicate experiments.

## 5.1 Hyperparameter Settings

In our experiments, we extensively evaluate the performance of the model under varying hyperparameter settings. We searched for the optimal coefficients of Laplacian regularization terms within the set $\{0.001, 0.01, 0.1, 1, 10, 100, 1000\}$, and a coefficient of 100 was selected based on the achieved accuracy.

We employ various kernel functions to assess the degree of closeness between two samples, facilitating the generation of similarity matrices within the Laplacian regularization. We conducted a series of experiments employing various kernel functions—namely Gaussian, Matérn, and Neural Network kernels—to evaluate their effectiveness within our optimization method. Prior to computing the similarity matrices, we normalize the samples by their mean square deviation. Upon conducting a comprehensive empirical analysis and undergoing a meticulous validation process, we concluded that the Gaussian kernel function in Section 3.4 is the optimal choice. The parameter $\sigma$ was determined to be $\sqrt{n-1}$ through extensive trials and validations, where $n$ represents the number of samples. Subsequently, this function was utilized in all following optimization effect tests unless stated otherwise. Other ancillary parameters were retained at their default values as specified in the Python Optimal Transport (POT) library.

## 5.2 Comparative Analysis of Hutchinson's Method Variants

In the experiments conducted under the parameter settings described in the previous section, we meticulously investigated the accuracy and time performance of the original Hutchinson algorithm (denoted as Hutch) and its variant, Hutch++. These algorithms were tested with two types of randomly generated vectors: those with i.i.d. standard Gaussian-distributed entries (denoted **HutchG** and **Hutch++G**), and those with i.i.d. $\{+1, -1\}$ entries (denoted **HutchU** and **Hutch++U**), where the number of queries was uniformly set to **8**.

Our analysis included a comparison of these Hutchinson-based methods with a control group (**CG**), representing traditional methods that do not incorporate the Hutchinson algorithm or its variants. The detailed results of this comparative analysis are presented in Table 1.

Table 1: Comparative analysis of algorithmic performance.

|              | CG     | HutchG | HutchU | Hutch++G | Hutch++U | TrSVD-8 | TrSVD-64 |
|--------------|--------|--------|--------|----------|----------|---------|----------|
| Accuracy (%) | 67.80  | 64.17  | 63.97  | 64.72    | 64.30    | 67.60   | 67.33    |
| Std (Acc.)   | 0.00   | 4.29   | 3.55   | 2.70     | 2.83     | 0.00    | 0.00     |
| Time (s)     | 501.54 | 234.38 | 236.23 | 237.37   | 240.47   | 739.64  | 752.52   |
| Std (Time)   | 16.88  | 84.09  | 67.71  | 71.95    | 70.68    | 17.30   | 15.54    |

Moreover, we compared Hutchinson's method with the truncated-SVD method, contrasting the stochastic approach of the former: in Equation (2), the matrix $X_t$ has its dimensions reduced using the truncated Singular Value Decomposition (SVD), ensuring that its rank does not exceed $k$. The results corresponding to $k = 8$ and $k = 64$ are denoted as **TrSVD-8** and **TrSVD-64**, respectively, in Table 1.

As seen from Table 1, methods underpinned by the Hutchinson approach can achieve an accuracy rate that deviates less than 4% from the traditional method, all while spending less than half the time. The four Hutchinson-inspired methodologies exhibit comparable temporal and accuracy metrics. Yet, the Hutch++ variants distinctly outperform their counterparts in terms of consistency, giving superior control over the standard deviation of the accuracy, which matches our theoretical expectations. Consequently, we use the Hutch++U variant for the subsequent experimental evaluations, and for the sake of clarity and conciseness, we will refer to it simply as **Hutch++**. We can see that by increasing the number of matrix-vector queries in the trace estimator, we naturally obtain a more precise estimator. In the next section, we embark on a granular exploration, showing how variations in the number of queries impinge upon both the efficiency and accuracy of Hutchinson-based methods.

## 5.3 ANALYSIS OF PARAMETER TUNABILITY: COMPUTATIONAL TIME AND ACCURACY

The number of queries in Hutchinson's method plays a significant role in both computational time and accuracy. With the previous parameter settings, and opting for the Gaussian kernel function for our experiments, we observe the following. Table 2 selects certain instances of the number of queries, with more extensive data depicted in Figure 1, to illustrate the relationship between the accuracy, the computational time, and the number of queries in Hutch++ method. Given the sample size in our experiment (1500), the number of queries must be set to a relatively small value to accurately demonstrate the advantages in time complexity.

In terms of accuracy, a small number of queries (fewer than 8) significantly impairs the accuracy of OT. However, with a slight increase in the number of samples, the accuracy under the op-

Table 2: Comparative analysis of Hutch++ performance with varying number of queries.

| Number of Queries | Accuracy (%)     | Time (s)             |
|-------------------|------------------|----------------------|
| Control Group     | $67.80 \pm 0.00$ | $501.54 \pm 16.88$   |
| 4                 | $62.19 \pm 5.18$ | $196.01 \pm 54.76$   |
| 8                 | $64.30 \pm 2.83$ | $240.47 \pm 70.68$   |
| 16                | $65.93 \pm 1.68$ | $325.86 \pm 100.43$  |
| 20                | $66.50 \pm 1.36$ | $375.19 \pm 110.02$  |
| 24                | $67.02 \pm 1.39$ | $427.47 \pm 133.87$  |
| 28                | $67.46 \pm 1.00$ | $496.61 \pm 148.73$  |
| 32                | $67.74 \pm 0.87$ | $505.22 \pm 164.49$  |
| 40                | $68.06 \pm 0.65$ | $623.88 \pm 220.19$  |
| 56                | $68.07 \pm 0.49$ | $791.81 \pm 249.76$  |
| 64                | $67.96 \pm 0.49$ | $826.16 \pm 227.41$  |
| 128               | $67.72 \pm 0.29$ | $1754.40 \pm 480.85$ |

timization of the Hutchinson algorithm markedly improves, as depicted in Figure 1. It is noteworthy that the accuracy peaks and slightly surpasses the accuracy of the conventional algorithm around a number of queries 48, after which it gradually declines to the accuracy level of the conventional algorithm. This can be interpreted as the stochastic perturbations introduced by Hutchinson's method rendering the algorithm

less prone to overfitting. With the increase in the number of queries, the standard deviation diminishes, ultimately converging to that of the conventional algorithm. Meanwhile, the computation time exhibits linear growth with respect to the number of queries, as depicted in Figure 1 (b).

As illustrated in Figure 1 (a) and (b), we observe a highly precise trade-off between time and accuracy. From the curves in the figure, we calculate that the Hutch++ method can achieve an average accuracy within 1% of the original algorithm in 80% of the time of the original algorithm, or attain an average accuracy within 3.25% of the original algorithm in 50% of the time of the original algorithm.

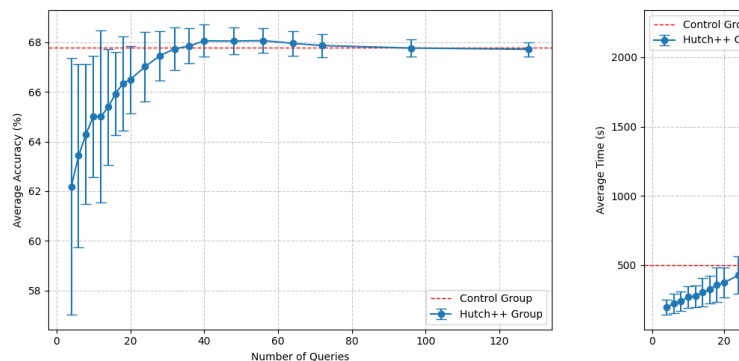
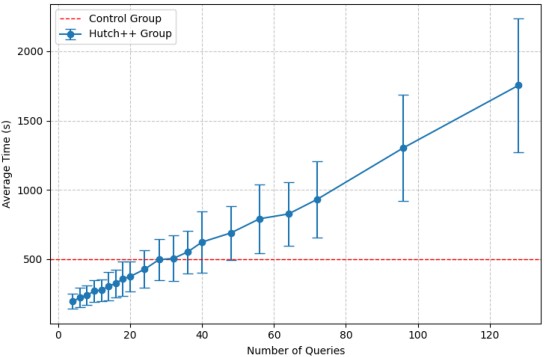

(a) Accuracy as a function of query number.    (b) Computational time as a function of query number.

Figure 1: (a) The relation between the average accuracy and the number of queries for the Hutch++ Group and Control Group. (b) The relation between the average time taken and the number of queries for the Hutch++ Group and Control Group.

Furthermore, our experiments substantiate that the enhanced computational efficiency of our Hutchinson algorithm maintains consistent effectiveness with variations in dataset size. Table 3 depicts the accuracy and time corresponding to the number of samples. In the scope of our experiment, both the source and target domains maintain an equivalent sample count, with the number of queries set to 8 in all experiments.

Table 3: The impact of dataset size on the accuracy and time of Hutch++ and Control Group algorithms.

| | Hutch++ | | Control Group | |
|---|---|---|---|---|
| Size | Accuracy (%) | Time (s) | Accuracy (%) | Time (s) |
| 1000 | $62.36 \pm 3.77$ | $121.02 \pm 37.52$ | $65.90 \pm 0.00$ | $282.58 \pm 11.17$ |
| 1250 | $63.39 \pm 4.17$ | $180.76 \pm 60.48$ | $67.36 \pm 0.00$ | $384.85 \pm 13.95$ |
| 1500 | $64.30 \pm 2.83$ | $240.47 \pm 70.68$ | $67.80 \pm 0.00$ | $501.54 \pm 16.88$ |
| 1750 | $64.46 \pm 3.56$ | $315.31 \pm 87.46$ | $68.51 \pm 0.00$ | $652.99 \pm 14.90$ |
| 2000 | $65.81 \pm 4.04$ | $435.55 \pm 143.33$ | $69.20 \pm 0.00$ | $815.31 \pm 20.73$ |
| 2250 | $66.32 \pm 2.47$ | $699.21 \pm 236.96$ | $69.02 \pm 0.00$ | $1132.16 \pm 15.43$ |
| 2500 | $64.51 \pm 5.26$ | $652.49 \pm 212.49$ | $67.76 \pm 0.00$ | $1200.00 \pm 18.86$ |

## 5.4 ROBUSTNESS TESTING UNDER SUBOPTIMAL CONDITIONS

In our series of experiments, we observed a considerable sensitivity of traditional algorithms to parameter variations, with minor perturbations in parameters leading to notable impacts on both accuracy and com-

putation time. Due to its inherent stochastic characteristics, our Hutchinson-based method demonstrated significant superiority under certain suboptimal parameter settings, compared to traditional algorithms.

The maximum number of iterations acts as the internal iteration limit for the conditional gradient solver. The default value in the POT library is set to $100,000$, to ensure sufficient convergence. Reducing this value to $1/10$ of the default facilitates more manageable computational times at the expense of accuracy. In such circumstances, the evident advantages of the Hutchinson method in terms of accuracy, compared to the conventional methods, become particularly pronounced. For instance, with an insufficient number of iterations, where the maximum iteration number was set to $\frac{1}{10}$ of a default value, we examined the accuracy and time performance of Hutch++ against traditional algorithms with a different number of queries, as presented in Table 4. Here, we have employed an alternative implementation of the Hutchinson algorithm to observe potential nuances in performance when the number of queries is less than four. The superior accuracy and time efficiency of the Hutchinson-based methods were clear.

Table 4: Performance of Hutchinson's Method with Gauss Kernel and 3-NN under Insufficient Iterations.

| Number of Queries | Gauss Kernel | | 3-NN | |
| --- | --- | --- | --- | --- |
| | Accuracy (%) | Time (s) | Accuracy (%) | Time (s) |
| Control Group | $57.53 \pm 0.00$ | $248.68 \pm 14.60$ | $29.00 \pm 0.00$ | $154.08 \pm 9.26$ |
| 1 | $49.00 \pm 12.26$ | $164.58 \pm 55.72$ | – | – |
| 2 | $57.35 \pm 7.71$ | $206.68 \pm 65.50$ | – | – |
| 3 | $57.61 \pm 7.54$ | $219.86 \pm 75.45$ | – | – |
| 4 | $58.04 \pm 7.53$ | $276.58 \pm 93.66$ | $38.53 \pm 4.22$ | $71.47 \pm 88.56$ |
| 8 | $60.83 \pm 4.88$ | $365.06 \pm 129.36$ | $38.73 \pm 3.98$ | $114.19 \pm 152.63$ |
| 16 | $61.39 \pm 4.59$ | $539.31 \pm 166.35$ | $38.47 \pm 4.09$ | $178.13 \pm 219.66$ |
| 32 | $61.12 \pm 6.00$ | $902.52 \pm 302.61$ | $36.38 \pm 5.05$ | $243.47 \pm 351.26$ |
| 64 | $61.94 \pm 5.77$ | $1657.36 \pm 525.62$ | $35.01 \pm 5.43$ | $661.45 \pm 794.15$ |

Previously, we employed the Gaussian kernel function to generate similarity matrices. An alternative, more coarse but highly practical approach to characterizing sample similarity is the k-NN method. A sample is deemed similar only if it is within the nearest neighbors of the other, leading to similarity matrices typically composed of 0, 0.5, or 1. In this scenario, a substantial amount of similarity information has also been lost. Under 3-NN conditions and the same suboptimal iteration setting mentioned, the results are depicted in Table 4, to be compared with the Gauss kernel case on the left. Hutchinson's method demonstrates precision substantially surpassing that of traditional algorithms, along with reduced computation time, showcasing its capability against adversarial perturbations.

It is noteworthy that the similarity matrices generated by 3-NN are notably sparse. Viewed from another perspective, we employed a stochastic algorithm for graph sparsification developed by Spielman & Srivastava (2011) and investigated the effect of sparsifying the matrix $L_s$ in Equation (2) produced by the Gaussian kernel function, simulating a process of information loss. In this experiment, the number of queries is set to 3. The results under suboptimal iterations are illustrated in Figure 2. It was observed that, during this process of information loss, the accuracy degradation of Hutchinson-based methods was markedly less than that of conventional algorithms. In the majority of cases, the Hutchinson method surpasses the original algorithm in terms of accuracy and computation time.

## 5.5 Comparison with Peer Algorithms

In alignment with the work presented by Courty et al. (2017), the accuracy of the OT-Laplacian method in the adaptation task from MNIST to USPS is reported to be $64.72\%$. It is noteworthy that their experimental setup differed slightly in terms of the sample size, utilizing 2000 samples from the MNIST dataset and 1800

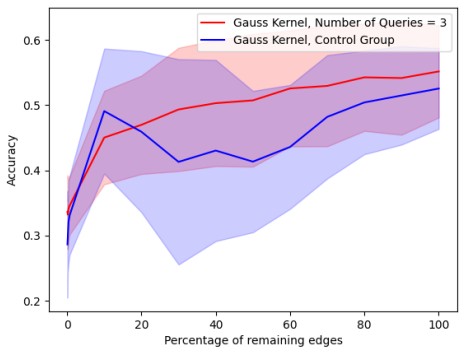

(a) Accuracy as a function of sparsity level.

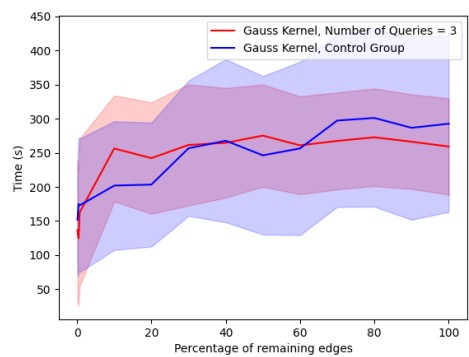

(b) Computation time as a function of sparsity level.

Figure 2: (a) The relationship between the accuracy and sparsity level under different similarity portraying methods. (b) The relationship between the computational time and sparsity level under different similarity portraying methods. The semi-transparent region represents the range of one standard deviation.

from the USPS dataset, compared to our uniform selection of 1500 samples from each. Apart from this difference in sample size, the methodologies for sample processing were consistent.

In experiments with equivalent sample sizes as Courty et al. (2017) under our settings, our Hutch++ method with 8 queries manifested an accuracy of $65.15\%$, with the control group achieving $69.67\%$, both surpassing the accuracy attained by their OT-Laplacian method. This level of accuracy is comparable to the highest accuracy observed in their study, achieved by their OT-GL method, recorded at $69.96\%$.

With equivalent sample sizes utilized in our study and comparable parameter settings applied, the average computational time for the OT-GL method in our tests was 1723.21 seconds, representing a substantial increase in comparison to the computational time required by our Hutchinson-based method, even when the query number is set to 40. Furthermore, in this scenario, the accuracy attained by OT-GL could not match that of our method. This underscores the improved computational efficiency of our algorithm relative to other methods with comparable levels of accuracy.

## 6 CONCLUSION

This research conducts a rigorous investigation into the applications of Hutchinson's method in Laplacian regularized OT problems, elucidating their efficacy and adaptability in DA tasks. The empirical validations confirm that our enhanced method significantly reduces the computational burden, achieving comparable accuracy, and thus establishing its capability to handle large-scale data in diverse DA settings. The Hutchinson-based OT method allows for a highly sophisticated trade-off between time and accuracy, by linearly adjusting computational time via the number of queries. The incorporated stochastic perturbations serve to mitigate overfitting, with the potential to enhance accuracy when the number of queries is appropriately selected. The method's robustness and adaptability under suboptimal conditions emphasize its resilience against various real-world challenges. However, there is an ongoing necessity to refine these stochastic methods to optimize accuracy and control variability in outcomes further. Furthermore, exploring the applicability of such methods beyond DA in other OT applications remains pivotal. These continual refinements and explorations are vital for advancing innovations and addressing the evolving challenges in the face of increasing volumes of data and diversifying application scenarios.

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
