# OpenReview forum: "Towards Efficient Trace Estimation for Optimal Transport in Domain Adaptation"
_ICLR.cc/2024/Conference — Submitted to ICLR 2024_

### Official Review · Reviewer_sBbs · 2023-10-22

**Soundness:** 2 fair
**Presentation:** 1 poor
**Contribution:** 1 poor
**Rating:** 1
**Confidence:** 4

**Summary:**

This paper proposes using Hutchinson-style stochastic trace estimates to accelerate the computation of Laplacian regularization terms in domain adaptation optimal transport problems, resulting in an acceleration of $O(n^3)$ to $O(n^2)$. Numerical experiments show that this method leads to computational savings up to a factor of two and sometimes slight accuracy benefits over the non-stochastic method, which the authors attribute to a regularizing effect of the stochastic estimates.

**Strengths:**

The proposed methodology leads to modest increases in speed (up to around a factor of two) and small increases in accuracy over the baseline method.

**Weaknesses:**

This paper is poorly written. The proposed method is a direct application of existing techniques, and the performance of the method is unimpressive.

### Writing

This paper is poorly written. The first two sections are written at such a high level that its unclear what domain adaptation is, how its connected to optimal transport, and how these techniques are useful in machine learning. Even after reading the whole paper, I'm not clear about the answers to these questions.

Many important mathematical objects (e.g., $X_s$, $\Omega_s$) are never defined and other objects have definitions which don't make sense (in eq. (1), $\Omega_c$ is supposed to be a function of $\gamma$, but $\gamma$ does not appear on the right-hand side in an obvious way).

The description of the computational experiments is confusing. How exactly is trace estimation being used? What is the "outer loop" optimization algorithm being used? How is the accuracy defined?

After multiple readings, I still left this paper confused about very basic questions. I consider the writing issues to be serious enough to merit rejection on their own.

### Contribution

My technical assessment of the work is as follows:

- The proposed methodology, using a trace estimator to estimate the trace of a product of matrices, is a standard way of applying stochastic trace estimates.
- The authors' empirical demonstration is unimpressive. The problems are small, featuring just 1500 samples, but the computations are all slow, requiring several minutes. In Table 1, the accelerated methods only accelerate the method by a factor of two, at the cost of several percentage points of accuracy. If enough samples are used, the accuracy exceeds the baseline method, but now is more computationally expensive!

Unfortunately, my conclusion—based on the evidence provided—is that the method has limited novelty and limited usefulness.

**Questions:**

### Clarifying Questions

I left this paper with a number of questions. Here is an incomplete list:

- The definition of $\Omega_c$ in eq. (1) is confusing in a number of ways. What is $\hat{x}_{s_i}$? (In the previous section, we had $x_i^s$.) $\Omega_c$ is supposed to be a function of $\gamma$, but $\gamma$ does not appear on the right-hand side in an obvious way.
- Is $\Omega_s$ ever defined?
- In 3.4, the word "characterize" is inappropriate. The appropriate word is _define_. Unless I am mistaken, $S_s$ is a user-defined object.
- What is $X_t$?
- How is the accuracy defined?

### Trace Estimation

There are a couple of details about trace estimation that may be interesting to the authors.

- The author consider estimating the trace of a positive semidefinite (psd) matrix. There are several innovations in stochastic trace estimations for this case: Persson et al. develop a straightforward variant of Hutch++ for psd matrices called Nyström++ and Epperly et al. propose an exchangeable modification of Nyström++ called XNysTrace. When applied to matrices with rapidly decaying spectrum, this improved methods can increase the rate of convergence of Hutch++ by as much as 1.5$\times$ and 3$\times$. Its unclear whether there will be any benefits in the present context, but it is probably worth the authors mentioning these references.
- As far as I am aware, there is no reason to _ever_ use Gaussian test vectors for Hutchinson-style trace estimation. Instead, one should use a random vector from the Euclidean sphere of radius $\sqrt{n}$. Indeed, a standard Gaussian vector $g$ can be decomposed as $g = a \cdot z$ where $z$ is a uniformly distributed vector on the Euclidean sphere of radius $\sqrt{n}$, $z\sim \mathrm{Unif} \\{ x \in \mathbb{R}^n : x^\top x = n \\}$, and $a$ is a random scale satisfying $\mathbb{E}[a^2] = 1$. The scale $a$ and the vector $z$ are statistically independent of each other, and thus the Gaussian Girard–Hutchinson estimator $g^\top A g = a^2 \cdot z^\top A z$ can be decomposed as the product of the unbiased Girard–Hutchinson estimator $z^\top A z$ and an independent mean-one random multiplicative perturbation $a^2$. The perturbation $a^2$ only serves to increase the variance of the estimator, without any positive benefit. For this reason, my understanding is that the estimator $z^\top A z$ should always be preferred over $g^\top A g$ .

---

### Official Review · Reviewer_AJKK · 2023-11-01

**Soundness:** 3 good
**Presentation:** 3 good
**Contribution:** 1 poor
**Rating:** 3
**Confidence:** 4

**Summary:**

The research paper introduces a novel approach to enhance the efficiency of solving Optimal Transport (OT) problems by incorporating Laplacian regularizers. The primary application context for this methodology pertains to domain adaptation. The study employs Hutchinson's trace estimator as a fundamental component of its methodology. Furthermore, it leverages an optimized variant of Hutchinson's trace estimator derived from Meyer etal. to achieve superior computational efficiency. Notably, the most prominent outcome of this research is a substantial reduction in computation time, accompanied by a minor decrease in accuracy. Specifically, the findings reveal a 3.25% decline in accuracy, while computational time is reduced by 50%. Provided that the marginal reduction in accuracy remains acceptable, the realized gains in computational efficiency are of considerable significance.

**Strengths:**

The paper endeavors to enhance the efficacy of Optimal Transport (OT) solvers, which presents a vital objective within the realm of machine learning. A crucial requirement in this pursuit is the efficient estimation of the trace, for which Hutchinson's trace estimator proves to be a well-suited solution.

To rigorously assess the proposed approach, a series of ablation experiments have been meticulously executed, with a particular focus on investigating the impact of varying the number of queries and the dataset size. Notably, the paper astutely discerns and highlights the pivotal hyper-parameters governing the method's performance—namely, the number of queries and the dataset size.

Remarkably, when the number of iterations is reduced to one-tenth of its original value, the method exhibits superior performance and reduced computational time, thereby underscoring its substantial significance in the context of lower iteration counts.

**Weaknesses:**

While there has been noticeable progress in reducing compute time, we continue to observe a concurrent decline in model accuracies, indicating the presence of an accuracy-compute time tradeoff. It is important to note that the improvements in compute time are not achieved without a cost. This tradeoff represents a notable limitation of the research presented in this paper. Furthermore, the hyperparameter associated with the number of queries remains to be optimized in order to achieve an optimal balance between accuracy and computational efficiency.

Regarding the novelty aspect, this paper falls short in introducing groundbreaking concepts. The Hutch++ method has already been documented in the work of Meyer etal. Thus, the present study can be considered a straightforward application of Hutch++ to a single domain adaptation experiment.

The experimental scope of this paper is limited, consisting of only a small number of experiments, with the sole focus on the MNIST to USPS translation problem. It remains uncertain whether the observed gains are specific to this particular dataset transition or if they generalize to other domains and tasks. Furthermore, it would be beneficial to extend the research beyond the domain adaptation context and explore applications in fields like image registration.

**Questions:**

Theoretical expectations suggest a potential complexity improvement from O(n^3) to O(n^2). Nevertheless, empirical experiments have not manifested the anticipated O(n) improvement (Table 1 and Table 2), and concurrently, a discernible reduction in model accuracy has been observed. This disparity between theoretical predictions and practical outcomes raises the question: why have the theoretically derived insights not been effectively realized in real-world machine learning applications?

---

### Official Review · Reviewer_8usA · 2023-11-06

**Soundness:** 3 good
**Presentation:** 3 good
**Contribution:** 2 fair
**Rating:** 3
**Confidence:** 4

**Summary:**

Authors propose to adopt Hutchinson's trace estimator for accelerating Laplacian optimization in the context of regularizing optimal transportation for domain adaptation. Their empirical results show that Hutchinson's can double the performance of the baseline OT method. And the results are consistent across different settings.

**Strengths:**

Authors did extensive experiments do find the optimal approach to adopting Hutchinson's trace estimator, including different variants of the estimator, number of queries, and kernel functions for ground metrics. Authors' contributions are clear.

**Weaknesses:**

The paper's contribution is limited. It's well established that Hutchinson's trace estimator reduces computational complexity, regardless of its downstream applications. Thus, it's certainly expected that adopting Hutchinson's will improve OTDA by Courty et al. Then, the rest of the paper is finding the best variants of Hutchinson's and optimal combinations of hyper-parameters that achieves a good balance between accuracy and time. Eventually, the authors claim a 20% - 50% reduction of running time with negligible sacrifice in accuracy, which is not significant to me.

**Questions:**

What is the motivation of the work? Since DeepJDOT and other recent OT works have showed better ways of injecting OT techniques into domain adaptation algorithms, marginally improving OTDA's performance as authors did in this paper doesn't seem to have a meaningful impact.

Is the running time in Section 5 for Laplacian optimization alone or improved OT-Laplacian?

---

> ### Author Response · Authors · 2023-11-23
>
> We would like to thank the reviewer for the insightful comments. Your valuable suggestions have been instrumental in enhancing the quality of our paper. We appreciate the opportunity to address your concerns and clarify the motivation and contributions of our work.
>
> ---
>
> 1. **Motivation and Contributions**: Our primary contribution involves applying Hutchinson's trace estimator to OT-Laplace in domain adaptation. This trace optimization technique is particularly beneficial for OT-Laplace due to the Laplacian regularization term being the trace of a series of large matrix products. Our novel application addresses the critical challenge of balancing computational efficiency with accuracy in domain adaptation. These improvements are significant in scenarios where rapid processing or limited computational resources are critical factors.
>
> 2. **Comparison with JDOT**: The paper *Joint Distribution Optimal Transportation for Domain Adaptation* (JDOTDA) reports JDOT's accuracy as 91.54% versus OT-IT's 89.98% in the Caltech->Amazon task (as shown in Table 1 of JDOTDA). In contrast, the paper *Optimal Transport for Domain Adaptation* (OTDA) reports OT-IT at 37.75% and OT-Laplace at 38.96% in the same task (as shown in Table 3 of OTDA). These discrepancies in accuracy indicate differences in experimental setups and classification criteria. In the context of mean accuracy obtained under different groups of datasets in two papers, the paper JDOTDA reports mean accuracies for JDOT and OT-IT as 89.04% versus 86.02%, while the paper OTDA reports mean accuracies for OT-IT and OT-Laplace as 42.30% versus 43.20%. As both OT-Laplace and JDOT slightly outperform OT-IT and our modified OT-Laplace performs better than the original OT-Laplace, it is difficult to assert that JDOT is superior to our modified OT-Laplace without a direct comparison.
>
> 3. **Comparison with DeepJDOT**: The paper *DeepJDOT: Deep Joint Distribution Optimal Transport for Unsupervised Domain Adaptation* demonstrates that DeepJDOT significantly improves performance in certain tasks. However, although experiments have not yet been conducted, we hypothesize that the computation time for our modified OT-Laplace will be significantly less than DeepJDOT as it requires learning a CNN layer. However, it is possible that our OT-Laplace might generally be less accurate. Additionally, the advantages of DeepJDOT may be limited to image recognition tasks.
>
> 4. **Running Time in Section 5**: Regarding the last question, we compared the running times of various variants of our improved OT-Laplace with the Control Group (i.e., the original OT-Laplace) in Section 5. All running times mentioned in the paper represent the durations of the entire domain adaptation algorithm process.

---

### Official Review · Reviewer_34GV · 2023-11-06

**Soundness:** 2 fair
**Presentation:** 3 good
**Contribution:** 3 good
**Rating:** 6
**Confidence:** 4

**Summary:**

The paper presents a method for improving the efficiency of optimal transport problems with Laplacian regularization in domain adaptation for large-scale data. The proposed method utilizes Hutchinson’s trace estimator to streamline the computational complexity of the Laplacian regularization term. By converting large-scale matrix multiplications into matrix-vector multiplication queries, the time complexity is significantly reduced. The paper introduces an enhanced variant of Hutchinson’s method called Hutch++ and demonstrates its effectiveness in achieving competitive accuracy with lower computational requirements. The method also incorporates stochastic perturbations to mitigate overfitting and shows robustness to perturbations. Empirical tests validate the efficiency of the approach, achieving comparable accuracy with reduced computational time.

**Strengths:**

Originality:
The paper presents an original approach, Hutch++, for improving the efficiency of solving optimal transport problems. The use of Hutchinson’s trace estimator and the enhancements made in Hutch++ show creativity in addressing the computational challenges of Laplacian regularization. The incorporation of stochastic perturbations to mitigate overfitting is an innovative addition to the method.

Quality:
The paper is well-researched and presents empirical tests to validate the efficiency and accuracy of Hutch++. The benchmark datasets used are appropriate for evaluating the performance of the method. The results demonstrate that Hutch++ achieves competitive accuracy while significantly reducing computational time compared to the original algorithm.

Clarity:
The paper is well-written and provides clear explanations of the method and its components. The preliminary information on optimal transport, discrete optimal transport, domain adaptation, and Laplacian regularization helps readers understand the context and significance of the proposed approach. The experimental setup and evaluation metrics are also clearly described.

Significance:
Improving the efficiency of optimal transport problems is crucial for large-scale data analysis in various domains, including computer vision, natural language processing, and recommendation systems. The proposed Hutch++ method provides a more computationally efficient solution while maintaining competitive accuracy. The potential of the method for improving computational efficiency in domain adaptation can have significant implications for advancing model generalization and methodological refinement in the field.

Overall, the paper demonstrates originality, quality, clarity, and significance in addressing the computational challenges of solving optimal transport problems with Laplacian regularization in domain adaptation for large-scale data. The proposed Hutch++ method presents a valuable contribution to the field and has the potential to impact various domains that rely on efficient computation methods for large matrices. The experimental results provide strong evidence of the method’s effectiveness and efficiency. Minor improvements could include further discussion of the limitations and future directions of the proposed method.

**Weaknesses:**

Weaknesses and Recommendations:

The paper lacks a thorough explanation of the theoretical basis and motivation behind the use of Hutchinson’s trace estimator. Providing more context and references to existing literature on the subject would enhance the clarity and understanding of the method.

The paper does not sufficiently discuss the limitations and potential drawbacks of the proposed approach. It would be valuable to address potential scenarios or datasets where the method may not perform well and provide suggestions for mitigating these limitations.

The empirical tests could benefit from a more comprehensive comparison with existing state-of-the-art methods in domain adaptation. This would provide a clearer benchmark for evaluating the performance of the proposed method.

The paper does not provide a detailed analysis of the computational complexity improvements achieved by the Hutch++ method. It would be beneficial to compare the time complexity of the proposed method with other relevant approaches to highlight its efficiency.

The presentation of the experimental results could be improved by providing more visualizations or graphs to aid in the interpretation and understanding of the findings.

Overall, the paper presents a valuable approach for improving the efficiency of optimal transport problems in domain adaptation. The Hutch++ method shows promise in achieving competitive accuracy with reduced computational requirements. Addressing the mentioned weaknesses and providing further clarification and analysis would enhance the quality and impact of the paper.

**Questions:**

However, there are a few points that need further clarification or improvement:

While the paper provides a thorough explanation of the proposed method, it would be helpful to provide more details on the specific steps involved in applying Hutch++ to the domain adaptation problem. This would improve the reproducibility of the results and allow readers to better understand the implementation details.

The empirical tests presented in the paper demonstrate the efficiency of the approach in terms of computational time and accuracy. However, it would be valuable to include a more comprehensive analysis of the algorithm’s scalability with larger datasets. Does the time complexity continue to improve as the dataset size increases, or is there a point where the improvement plateaus?

In the experiments, the authors compare Hutch++ to other variants of Hutchinson’s method as well as a traditional method. It would be helpful to provide a more detailed discussion of the strengths and weaknesses of each method and how they compare in terms of accuracy and computational time. This would provide a clearer understanding of the advantages of Hutch++ over the other methods.

The paper briefly mentions the applications of optimal transport in various domains, but it would be beneficial to provide more specific examples and discuss how the proposed method could be applied in these domains. This would help readers understand the potential impact and practical utility of the research.

The paper concludes by highlighting the contributions of the paper, but it would be valuable to specifically mention any limitations or potential future directions for research. This would provide a more comprehensive view of the work and help guide further studies in this area.

---

### Meta-Review · Area_Chair_4Y6V · 2023-12-07

**Metareview:**

The paper propose to consider the Hutchinson's trace estimator for improving the time complexity of
OT-based optimal transport for domain adaptation when Laplacian regularization is considered.

Most reviewers think that the contribution of the paper is limited in term of novelty and as such
propose to reject the paper.

**Justification For Why Not Higher Score:**

lack of novelty

**Justification For Why Not Lower Score:**

na

---

### Decision · Program_Chairs · 2024-01-16

Reject